# Bi-Directionality between Physical Activity within School and Fundamental Movement Skills in School-Aged Students: A Cross-Lagged Study

**DOI:** 10.3390/ijerph19137624

**Published:** 2022-06-22

**Authors:** Shanshan Han, Bo Li, Shuqiao Meng, Yaxing Li, Wenxia Tong

**Affiliations:** 1Institute of Sports Science, Nantong University, Nantong 226019, China; paiqiuhanshanshan@163.com; 2Physical Education College, Yangzhou University, Yangzhou 225127, China; 007917@yzu.edu.cn (S.M.); 007753@yzu.edu.cn (W.T.); 3Physical Education College, Shangqiu University, Shangqiu 476000, China; star_lee86@163.com

**Keywords:** children, physical activity, cross-lag model, fundamental movement skills, quasi-experimentation design, test of gross motor development

## Abstract

Background: Evidence has indicated the health importance of fundamental movement skills (FMS) and physical activity (PA) in children and their relationships seems bidirectional. However, their bidirectional relationship has not yet been fully answered in the literature. Aim: This study sought to determine bidirectional relationship between FMS and PA in children using cross-lagged study design. Methods: A total of 183 second-level students (8.8 ± 1.1 years old) from three primary schools in Henan Province, China were selected as subjects. The average number of steps per school day was used as the amount of PA in the school environment; the third edition of the test of gross motor development was used for FMS testing. The baseline data (T1) and tracking data (T2) were collected at the beginning and end of the fall semester, respectively. The two tests were separated by 3 months (11 weeks), and a cross-lag model analysis was performed. Based on the hypothetical model, we tested the cross-lag effect of children’s PA and FMS. Results: The model fit index was χ^2^/df = 2.861 (*p* < 0.001, *n* = 183); goodness of fit index GFI = 0.900; NFI = 0.909; CFI = 0.931 and the 95%CI was between 0.071–0.192. The RMSEA = 0.063, and the standardized residual root mean square SRMR = 0.029. The T1 FMS can be used to predict the number of steps in the T2 teaching days with statistical significance (*β* = 0.22, 95% CI: 0.07–0.38, *p* = 0.003). However, the T1 steps cannot be used to predict the T2 FMS (*β* = 0.05, 95% CI: 0.07–0.13, *p* = 0.475). Further analysis shows that the main contributor to these relationships are ball skills in the FMS. Conclusions: The relationship between children’s fundamental movement skills and PA is not two-way. Students with higher FMS are expected to reach higher levels of PA after undergoing school PA in a teaching cycle. The PA of the students can be improved by improving their motor skills, which further improves their physical and mental health.

## 1. Background

In the past decade, obesity and physique decline in children and adolescents have become major problems not only in education and sports administrative departments, but also throughout China [1]. Physical inactivity has become an important factor in the decline of physical and mental health of children and adolescents in various countries [2]. Physical activity (PA) plays a positive role in controlling obesity, improving cardiorespiratory endurance, and reducing the risk of many chronic non-communicable diseases (such as hypertension, diabetes) in adulthood [3]. Promoting sports participation, improving PA, and reducing sedentary behaviors improve the health of children and adolescents, and have become a common demand for public health and physical education globally [4].

In 2016, the Chinese government issued the ***Opinions on Strengthening School Sports to Promote the All-round Development of Students’ Physical and Mental Health***, which specifically proposes that the training of sports skills should be taken as a guide to improve students’ exercise levels and further promote the physical and mental health and physical fitness of teenagers and children. In 2020, the Chinese government issued the ***Opinions on Comprehensively Strengthening and Improving School Physical Education in the New Era***, which clearly proposes the gradual improvement of the school physical education model of ***health knowledge + fundamental movement skills (FMS) + special sports skills***. The concept of FMS has gradually expanded from the academic research field to the educational practice level. Existing research also suggests that improving the level of motor skills of children and adolescents to improve PA may solve the current dilemma of PA improvement [5,6].

FMS are defined as basic learned movement patterns that occur naturally in the human body. This includes three lower categories: (1) locomotion skills (specific action forms, such as walking, running, jumping, and sliding); (2) object control skills (such as grasping, throwing, kicking, catching, hitting, and fighting); and (3) stability skills (such as spinning, turning, and bending) [7]. Childhood FMS are an important indicator for the assessing, diagnosing, and monitoring of individual motor development [8,9,10]. Considering the importance of FMS development, conducting a large-scale longitudinal survey of children’s FMS is a common practice in developed countries. For example, the USA evaluated children’s motor development as early as the 1980s and included motor development in American schools. The ***Report on the Status of Physical Education Development*** analyzes indicators and conducts follow-up surveys for this purpose [11]. The FMS of children aged 0–17 is an important aspect of the German national survey “German Children and Adolescent Health Interview and Research Project [12].” The ***mountain of motor development theory*** states that the stage of formation of children’s FMS is between the ages of 1–7, and the stage of proficiency is between the ages of 7–12. Only when children have mastered FMS during this period can they be guaranteed the ability to flexibly adapt to different sports and sporting environments, increase sports self-confidence, and increase their willingness to participate in physical exercise independently in future school life and lifelong sporting activities [13].

Stodden et al. established a dynamic mechanism model that affects the trajectory of children’s PA. This model sets the motor ability as one of the potential factors that affect an individual’s PA level and weight status for the first time; their study initially analyzed different ages. The core view of this model is that PA promotes athletic ability in childhood; however, in childhood, the opposite mechanism occurs, in which exercise ability promotes PA [14]. The results of the latest meta-analysis show that even though the current empirical research indicates that medium-to-high-intensity PA and all types of PA are significantly positively correlated with FMS in young children, the lack of longitudinal research evidence can neither support nor oppose Stodden’s proposal. Thus, further longitudinal research data are required to verify the two-way relationship between FMS and PA [15]. At present, when developing adolescent and child obesity intervention and health promotion policies in Western developed countries, one of the important directions is to improve the FMS level in childhood [16]. Most previous studies have explored the relationship between the two from the perspective of cross-sectional design; however, this can only explore the correlation between the two—the discussion on causality is weak [17]. Therefore, this study attempts to explore the two-way relationship between FMS and PA in children from the perspective of quasi-experimental design using the cross-lag model analysis theory. This study also aims to expand the perspective of sports research regarding youth and children and provide references for research on their health promotion.

## 2. Methods

### 2.1. Research Objective

Three primary schools were selected in Shangqiu Henan Province, China using the cluster sampling method. There were six administrative classes for grades three and four with a total of 199 students; each school chose one class each from grades three and four. To ensure the integrity and comprehensiveness of the sample selection, two of the three schools were located in the city, and one school in the countryside. Participation in the baseline test was utilized as the first test (T1), and the follow-up test was the second test (T2). A total of 189 students participated in the full study.

### 2.2. Research Design

A cross-lagged panel correlational design was adopted [18]. The methodology of the cross-lag design is to first obtain a number of correlation coefficients that vary over time and then determine which factors lead to the results based on the magnitude and direction of these correlation coefficients. According to the methodology of the design, two tests were conducted on the same test group at an interval of 3 months. According to ***the mountain of motor development theory***, children’s FMS change at a faster rate during the ages of 3–8 years old. From an empirical point of view, Burns explored the changes in FMS before and after the summer vacation. The results showed that for both boys and girls, the FMS of children, measured based on the second test of gross motor development (TGMD-2), showed significant differences before and after the summer vacation. Therefore, a regular teaching cycle (16 weeks) was selected in this study to meet the theoretical needs of constructing a cross-lag model [19]. Considering the actual needs of the test, the second week of school and the second or third week before the summer vacation were selected for tests T1 and T2, respectively, without affecting the normal teaching order.

In accordance with ethics requirements, the following tasks were completed before the test under the supervision of the ethics committee: (1) communicating with the school regarding the subjects and issuing informed consent forms for the subjects to the school legal person; (2) soliciting the opinions of the subjects’ guardians, and issuing and obtaining filled out informed consent forms for the subjects from the guardians; (3) the testers were trained before the first test on aspects including the test process implementation, emergency handling, tool use and recovery, and data recording; (4) before the formal test, the informed consent form for children under 12 years old was read aloud to the subjects; (5) the FMS test site personnel includes testers, physical education teachers in the administrative class, and doctors in the school. The cross-lag design measures and analyzes the relationship between two variables (A: number of steps, B: FMS) during T1 (2019.3.4~3.8), in which step counting is performed for four consecutive days. The FMS measurement was performed on one day; 197 valid datapoints were obtained for the first test (2 students were unable to complete some items in the FMS test). 

This part of the data was used as the baseline data source. The same measurement and analysis were conducted for T2 (2019.5.27–5.31), and 189 valid datapoints were obtained for the second test; this portion of the data was used as the source of tracking data. A total of 183 valid datapoints (from children aged 8.8 ± 1.1 years old) were obtained according to the school number before and after the test. Among them, 101 female students were included, which accounted for 55.2% of all students. When there is a significant difference between cross-lagged correlations, it can be inferred that there is a causal relationship between variables A and B [18]. To reduce the system error in this design, the same batch of testers used the same batch of test tools for T1 and T2.

### 2.3. Measuring Tools and Methods

(1)Physical activity

The number of steps taken by students in the school environment was selected as the amount of PA. Currently, there are many methods to measure PA [20]. Among them, wearable devices have attracted the attention of researchers, owing to their low price and high reliability. Wearable devices based on accelerometers (such as civil bracelets, mobile phone applications, and collars) are the mainstream paradigm of physical activity measurement from the current public health research perspective. Although bracelets or mobile phone applications display different data forms (such as the number of steps or energy consumption), the principles of data collection and calculation are the same or similar [21]. Therefore, the selection of the number of steps as the indication of PA in this study has a particular theoretical nature. Considering the project cycle and funding, as well as the simplicity and operability of the measurement, a particular brand of bracelet was selected as the measurement apparatus in this study, and it is currently available on the market (The relevant parameters of the bracelet can be obtained from http://www.mi.com/shouhuan3/specs (accessed on 1 May 2020)). The pedometer function of the bracelet uses a three-axis acceleration sensor. The three-axis acceleration sensor in the ring is currently considered to be the most reliable type of pedometer sensor [22]. The bracelets were worn for four consecutive teaching days between 8 am and 4 pm. They were worn by a fixed tester before school, and the data were collected and recorded after school. The bracelets were uniformly worn on the subjects’ right wrists. Each student only recorded the number of steps taken during the eight hours of school activities every day. The selected administrative class had two physical education classes per week, each 40 min in duration. Each bracelet had an independent identification number that contained the name and student number information of each student. Each time the bracelets were retrieved, their data were entered into Microsoft Excel by a fixed researcher.

(2)FMS test

Using the American test of gross motor development, the TGMD was developed by Ulrich of the University of Michigan. The test comprised children aged 3–10 years as the test subjects to evaluate the development level of children’s FMS. The evaluation used a combination of process evaluation and result evaluation by scoring the children’s completion of 13 gross movements, including forward sliding steps, and then evaluating the development of the test subject’s FMS. The latest version, TGMD-3, contains two dimensions: the locomotor subtest and ball-skills subtest, which correspond to movement skills and object control skills in children’s FMS. The TGMD-3 was released in 2013 and was used to conduct large-sample normative reliability and validity data testing. The TGMD-3 primarily includes the two subtests of movement and football skills. The displacement test includes six skills: run, gallop, hop, skip, horizontal jump, and slide. The ball skill test includes the two-hand strike of a stationary ball, one-hand forehand strike of a self-bounded stationary strike ball, one-hand stationary dribble, two-hand catch, kicking a stationary ball, overhand throw, and underhand throw. There were 13 test actions in the two subtests. Each test action had 3–5 specific completion standards for the test actions to be correct. A point was awarded for execution; however, zero points were given for a failure to execute the action correctly, and the test was repeated. The score range was 0–100, of which the total scores of the mobile test and ball skill test were 46 points and 54 points, respectively [23]. The TGMD exhibited good reliability and validity in China. Diao et al. conducted a validity test on TGMD-3, in which the α coefficient of the internal consistency reliability total scale was 0.81, the Pearson correlation coefficient of the total scale in the test–retest reliability was 0.974, the total inter-rater reliability, the Pearson correlation coefficient of the table was 0.983, and the confirmatory factor analysis in the structure validity was χ^2^ = 151.885, df = 64, χ^2^/df = 2.37. The NNFI value of the scale was approximately 0.95, the CFI and GFI values both exceeded 0.95, and the approximate error root mean square (RMSEA) and the standardized residual root-mean-square (SRMR) values were both less than 0.05. The scale had a satisfactory goodness of fit. Multi-group analysis showed that the tool was valid across gender and school types. It has good stability and validity and can be used as a tool to assess the development of FMS in children aged 3–10 years in China [24]. The TGMD series is currently one of the most widely used FMS evaluation tools worldwide [19]. In this study, the TGMD operating specifications were strictly followed. Before the implementation of the test, the testers conducted intensive learning based on the test video materials publicly released by the TGMD developer; they checked the specifications of the equipment, and produced a complete TGMD operating process. There were 12 testers in this study, all of whom were undergraduates majoring in physical education.

### 2.4. Statistical Methods

Prior to formal data analysis, all variables of interest in our study were tested for confirmation as normal distribution. The main variables used in the analysis included the average T1 and T2 steps on class days, the total T1 and T2 motor skills scores, and the T1 and T2 motor skills subtest scores. AMOS 25.0 was used to build the model and perform the cross-lag analysis. The cross-lag model can analyze the relationship between two or more observed variables measured at two or more different time points, including autoregressive effects (variables from the latter time point are related to themselves) and the cross-lag effect (related to the variables). The maximum likelihood method was used to test the fit of the model. The cross-lag analysis result report includes the standardized regression coefficient (*β*) and the corresponding 95% confidence interval (CI).

## 3. Results

Of all study participants (*n* = 183), the mean age was 8.8 years old, and boys accounted for 47.5% (*n* = 87) in the overall sample. Table 1 presents the descriptive statistics results of the two tests. After three months of school activities, the total PA of the same group of subjects decreased, and their FMS improved. With reference to the study by Wu that uses item parceling, the pre-test (T1) and post-test (T2) PA (number of steps) and the observed variables of the FMS were processed according to the corresponding rules [21].

Based on the hypothetical model, the cross-lag effect of children’s PA and FMS (Figure 1) was tested. The results were as follows: model fit index χ^2^/df = 2.861 (*p* < 0.001, *n* = 183); goodness of fit index GFI = 0.900; NFI = 0.909; CFI = 0.931 and the 95%CI was between 0.071–0.192. The approximate root-mean-square error RMSEA = 0.063, and the standardized residual root mean square SRMR = 0.029. The above data show that the constructed cross-lag relationship model has good fit and adaptability.

Figure 1 shows the results of the age-adjusted cross-lag model using the total sample. It can be observed from the data that the number of T1 steps is significantly correlated with T1 motor skills (*p* = 0.007); however, the number of T2 steps is not significantly correlated with T2 motor skills (*p* = 0.897). The T1 steps predicted the T2 steps with statistical significance (*p* <0.001), and the T1 motor skills predicted the T2 motor skills with statistical significance (*p* <0.001). The motor skills of T1 significantly predicted the number of steps in T2 (*p* = 0.003); however, the number of steps in T1 could not predict the motor skills of T2 (*p* = 0.475). According to Eisma [22], if the correlation between variable A (T1) and variable B (T2) is greater than that between variables B (T1) and A (T2), and simultaneously, the correlation between variable A’s T1 and T2 is greater than that between variable B’s T1 and T2, it can be inferred that there is a causal relationship between A and B and that A and B are the independent and dependent variables, respectively. This confirms that there is a causal relationship between children’s FMS and PA, and the former may be the reason for a change in the latter, i.e., children’s FMS may be the causal variable of PA.

Furthermore, FMS comprises movement skills, object control, and stability skills; therefore, to determine whether the score of a TGMD-3 subtest was related to the number of steps, a cross-lag model of the subtests and the number of steps was established. The data show that the total score of the T1 ball skills subtest can significantly predict the number of T2 steps (β = 0.22, 95% CI: 0.06–0.33, *p* < 0.001); however, the T1 mobility skills subtest cannot significantly predict the number of T2 steps (β = 0.07, 95% CI: −0.17–0.33, *p* = 0.773). Moreover, the statistically significant covariance observed between the T1 motor skills and the T1 steps using the TGMD-3 total score was only identified in the ball skills subtest scores (*p* < 0.001). However, it was not identified in the mobility subtest scores. Therefore, the relationship observed in the FMS total score in the cross-lag model is primarily owing to the ball skill subtest score.

## 4. Discussion

This study uses a cross-lag quasi-experimental design with a sample of level two students as subjects, and it verifies the two-way relationship between PA and FMS in a school environment in a regular teaching cycle. The corresponding results show that the overall FMS before the semester, especially ball skills, can significantly predict PA in the school environment after the semester; however, this is not a two-way relationship. The results of the study support the current theoretical basis for exercise intervention in the school environment to increase the amount of students’ PA. Burns’ research also uses the cross-lag model to explore the two-way correlation between FMS and PA before and after the summer vacation [25], which is consistent with the results of this study. However, this type of PA does not occur in the school environment, which shows that the promotion strategy of PA may be multilayered [6]. The above results also verified Bauman’s social ecology model of PA. The influencing factors of PA are not limited to the physical and psychological development of individuals; they are also affected by factors such as interpersonal communication, environment, and policies [26].

Furthermore, previous studies focus on the positive correlation between the control of FMS and children’s participation in moderate-to-vigorous physical activity (MVPA); however, there is no clear relationship with low- and micro-intensity PA. For example, Fisher et al. measured 15 FMS in children at approximately 4 years of age and recorded PA using an accelerometer. Their results showed that the correlation between the FMS score and children’s MVPA time (r = 0.18, *p* < 0.001) was highly significant, and it was related to low-intensity exercise time [5]. Wrotniak et al. also observed that FMS is positively correlated with the duration and activity of children’s MVPA and that it is negatively correlated with sedentary behavior [27]. Studies have recently determined the relationship between motor skills and PA through educational intervention or teaching experiments. A study conducted a controlled intervention experiment on 460 children who were randomly divided into groups. The study showed that after the intervention, the two groups of children exhibited FMS and PA. There were significant differences in daily MVPA [28], and children with high motor skill scores engaged in higher intensity PA [29]. This also provides a reference for follow-up interventions to increase the amount of PA for children. In view of the limitations of the PA measurement method in the present study, no stratified analysis of the strength of the PA was conducted.

For issues related to movement development, it is always difficult for results from cross-sectional studies to have strong external validity. Therefore, the typical approach is to design a series of quasi-experimental designs or longitudinal research designs [18]. Wang’s study showed that FMS can indeed predict the future PA status of children and adolescents, to a certain extent, from the results of quasi-experimental design. However, the published experiments have not fully confirmed that the development of FMS can improve the future PA level of children and adolescents. The current level of motor ability in children and adolescents may be a predictor of future PA levels [30]. Some studies used longitudinal research and sequential research methods to verify the prediction of FMS development in early childhood for future PA participation. Lopes et al. tracked 285 children, aged 6–10 years old, from 2002–2007 and observed their stability skill level at the age of 6. They observed that a large number of children have good stability skills. In the subsequent 3 years of their study, only a slight decrease in PA level was observed, whereas the PA of children with low and medium stability skill levels decreased more. Stability skills are an important dimension of the FMS [31]. Another study tracked 281 children, aged 6–11 years old, using a multiple linear regression model, and they demonstrated that FMS levels can predict future PA levels [32]. It can be observed that in the current related research on FMS and PA, the research results of quasi-experimental designs and longitudinal research designs are also consistent with the results of this study.

The results of this study do not support the existence of a two-way causal relationship between FMS and PA. However, they are consistent with the dynamic mechanism model proposed by Stodden, which affects the trajectory of children’s PA, i.e., children’s effective acquisition of FMS and rapid acquisition of FMS development can effectively promote PA, especially in elementary school [14]. The study by Burns used a research design similar to this study to measure the FMS and PA of 7–9-year-old children before and after summer vacation. The results of the cross-lag model analysis are consistent with those of this study [25]. A systematic review by Barnett et al. showed that among the many influencing factors of FMS, the association of PA to object control skills and even movement skills is uncertain, which is also consistent with the results of this study [33]. However, there are currently teaching experimental studies that intervene in children’s PA to improve FMS. For example, a study by Wang promoted gross motor development in children aged 4–5 years through multiple PA modules. The results showed that multiple PA modules can improve children’s object control skills and movement skills to promote the development of children’s gross movements. The theoretical basis of this study was that PA is the basic method to realize movement development [34]. Therefore, more empirical data are needed to study the two-way causality between the FMS and PA variables.

The results of this study also show that the relationship of predicting PA using FMS is dominated by the ball skills subtest in TGMD-3, which primarily reflects the object control skills in FMS. Movement and object control skills are very important subcategories of FMS, in addition to stability skills. Test comparisons by some studies have shown that object control skills are positively correlated with children’s PA [28], and children with high levels of object control skills are more willing to invest in MVPA [35]. Other studies have reported that children with high levels of displacement motor skills have significantly reduced their sedentary time, and the proportion of MVPA in these children is significantly higher than that of other children [36]. It is worth noting that existing studies have shown that children of different genders exhibit different FMS effects on PA participation. A study by Cliff proposed that object control skills are highly correlated with boys’ PA and MVPA, whereas displacement motor skills are correlated with girls. This may be related to the fact that girls are willing to participate in sports activities that require fewer object control skills [37]. Compared with the total motor skill scores, the relationship between the two motor skills subcategories of locomotion skills and object control skills with children’s PA has not reached a consensus. In our study, we found that locomotor skills at baseline could not predict later PA, which is a study finding different from previous studies. Some possible reasons could be applied to explain it. First, unlike ball skills, locomotor skills may not trigger motivation or interest of engaging in PA for children. Second, our study focused on PA within school environment; but owing to some conditions, we used daily steps to reflect PA levels, which may not reflect overall levels of PA. Taken together, in our study, it seems impossible that locomotor skills may not be associated with later PA. However, more studies should confirm our explanations. Moreover, the related effects of adding gender variables are not clear [38]. This study proposes that childhood is a period of rapid development of FMS, and skills of different dimensions are different at maturity time nodes. For example, the maturity of particular mobility skills occurs earlier than that for object control skills [39]. This difference is related to the environment in which the children are, which includes factors such as their culture. Therefore, the measurement results for different ages and regions may cause this difference.

The results of this study can enrich the theoretical basis of the understanding of the physical and mental health of children and adolescents in the field of public health. PA refers to any physical action produced by the skeletal muscles that consume energy. Physical inactivity (lack of PA) is considered the fourth leading risk factor for death worldwide (accounting for 6% of global deaths) [4]. Therefore, in the current public health field, promoting PA has become the focus of research. Furthermore, the factors that affect the PA of children and adolescents have become the key to ***promoting PA***. That is, the verification factors that affect the PA of children and adolescents can be used as a theoretical basis to improve PA. Among the many factors that affect PA, the development of FMS in childhood restricts the development of children’s lifelong sports to a certain extent. One of the main questions of this study is whether it is possible to promote PA from the perspective of promoting FMS. The results of this study provide the relationship between the two variables to an extent. The results also provide a theoretical basis for the development of physical and mental health in children and adolescents in the field of public health.

### Limitations

Through quasi-experimental research and cross-lag research design, this study examines the internal connection between children’s FMS and PA, which is helpful in demonstrating false relationships and clarifying real problems. However, FMS contains three dimensions (movement skills, object control skills, and stability skills), and the TGMD-3 selected in this study only contains two dimensions (movement skills and object control skills); this study does not discuss stability skills. The speed of physical development in childhood is relatively high. However, this study did not examine the relevant indicators of children’s physical development (such as BMI and step length), which will also affect the external validity of the research results to a certain extent. Additionally, some potential factors (e.g., daily activities during holidays) that may influence the association among PA and FMS or vice versa owing to the measurement restrictions. This would be a barrier to more accurately present the associations among PA and FMS. We should also admit that small sample size in our study would limit research generalizability. Although the cross-lag study can draw multivariate causal conclusions, it cannot prove complete cause and effect. It should be combined with cross-sectional empirical research and other means to further demonstrate this. PA measurement uses objective instruments, such as pedometers, which cannot distinguish the intensity of PA. Therefore, this study did not classify and analyze PA of different intensities. In the future, multiple long-term longitudinal tracking surveys should be conducted over years and combined with cross-sectional studies and improved measurement techniques to ensure the comprehensiveness and long-term stability of the research conclusions.

## 5. Conclusions

Children’s FMS (especially object control skills) can be used to predict their PA within school after 3 months (at the end of the semester). Students with higher FMS (especially object control skills) are expected to reach a higher level of PA within school after the teaching cycle of school PA. The results of this study also verify the dynamic mechanism that affects the trajectory of children’s PA within school. This is in agreement with the requirements of the motor skills teaching objectives in the current ***Compulsory Education Physical Education and Health Curriculum Standards (2011)*** in China. Motor skills reflect the basic characteristics of physical exercise. These can be used as the main method in the physical education and health curriculum and are an important component of curriculum learning. They can also be used as the main method to achieve other learning goals. The results of this study suggest that students’ PA can be improved by improving their motor skills, thereby further improving their physical and mental health. The results of this research can provide data support for basic education, physical education, and health curriculum reform.

## Figures and Tables

**Figure 1 ijerph-19-07624-f001:**
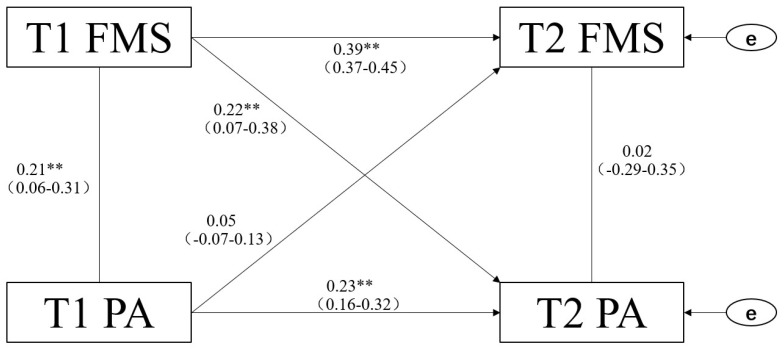
Cross-lag model of PA and FMS. ** denotes a statistically significant path coefficient *p* < 0.01.

**Table 1 ijerph-19-07624-t001:** List of descriptive statistics results (*n* = 183).

	T1 (Pre-Test)	T2 (Post-Test)
Number of steps	4451 ± 1618	4440 ± 1641
Fundamental movement skill	67.3 ± 12.7	76.1 ± 10.3
Locomotion skills	32.6 ± 8.9	32.9 ± 9.1
Object Control skills	35.6± 10.0	43.1 ± 9.2

## Data Availability

The datasets used and/or analyzed during this survey are available from the corresponding author upon reasonable request.

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
