# Peer review of "Bi-Directionality between Physical Activity within School and Fundamental Movement Skills in School-Aged Students: A Cross-Lagged Study"

_ijerph, 2022, doi:10.3390/ijerph19137624_

Round 1

Reviewer 1 Report

It is a well-written manuscript with good theoretical support. However, I have some suggestions that could improve it before publication

In the first place, since it is a linear technique, it is necessary to study the assumptions of this type of technique (normality, linearity...)   Second, since this is a quasi-experimental study in which there is no manipulation of independent variables, I would recommend that the authors think about possible extraneous variables that may influence the relationship between the variables and include them in the limitations, for example, BMI, the activities they did during the holidays, if it is a boy or a girl...  

Finally, I think it is important to discuss some results, such as: why the T1 mobility skills subtest cannot predict the number of T2 steps? Also is important that limitations include the lack of generalization of the results because of the sample type an size.

Author Response

Responses to Reviewer 1

It is a well-written manuscript with good theoretical support. However, I have some suggestions that could improve it before publication

Response: Thank you very much!

In the first place, since it is a linear technique, it is necessary to study the assumptions of this type of technique (normality, linearity...)   Second, since this is a quasi-experimental study in which there is no manipulation of independent variables, I would recommend that the authors think about possible extraneous variables that may influence the relationship between the variables and include them in the limitations, for example, BMI, the activities they did during the holidays, if it is a boy or a girl...

Response: Thank you very much! We have added relevant lines in the revised manuscript, please see the highlight in statistical analysis section (Line 204) and that in limitation section (Line 375 – 379).

Finally, I think it is important to discuss some results, such as: why the T1 mobility skills subtest cannot predict the number of T2 steps? Also is important that limitations include the lack of generalization of the results because of the sample type and size.

Response: Thank you very much! We have revised, please see Line 338 – 345 and Line 375 – 379.

Reviewer 2 Report

Dear authors, thank you very much for the opportunity to review your paper.

It is a research design that tries to answer a question that reminds me of the chicken and egg relationship. You examine the relationship between PA and FMS as the two only factors. Of course, there is a relationship between PA and FMS but it cannot be a direct one. PA and FMS are multidimensional. They depend on weight, height, BMI, gender, quality of the family environment, etc. None of these factors were examined, which may explain some of your results. You must take a step back and redesign your research. Not in this paper, but in others. 

There are no obvious mistakes, but this study needs more factors to be examined. 

Please, keep your main purpose: to increase PA in schools.

1. Abstract: Please change the first two sentences. Instead, write the purpose of your study.

2. 1 Research objective: Give more information about your sample (sex, age).

Author Response

Responses to Reviewer 2

Dear authors, thank you very much for the opportunity to review your paper. It is a research design that tries to answer a question that reminds me of the chicken and egg relationship. You examine the relationship between PA and FMS as the two only factors. Of course, there is a relationship between PA and FMS but it cannot be a direct one. PA and FMS are multidimensional. They depend on weight, height, BMI, gender, quality of the family environment, etc. None of these factors were examined, which may explain some of your results. You must take a step back and redesign your research. Not in this paper, but in others. There are no obvious mistakes, but this study needs more factors to be examined.

Please, keep your main purpose: to increase PA in schools.

Response: Thank you very much! We have revised throughout the manuscript where necessary.

Abstract: Please change the first two sentences. Instead, write the purpose of your study.

Response: Thank you very much! We have revised them, please see Line 9 – 14 in the revised manuscript.

Research objective: Give more information about your sample (sex, age).

Response: Thank you very much. We have provided information in the revised manuscript, please see Line 214 – 215 in the revised manuscript.